# Enhanced Vital Parameter Estimation Using Short-Range Radars with Advanced Motion Compensation and Super-Resolution Techniques

**DOI:** 10.3390/s24206765

**Published:** 2024-10-21

**Authors:** Sewon Yoon, Seungjae Baek, Inoh Choi, Soobum Kim, Bontae Koo, Youngseok Baek, Jooho Jung, Sanghong Park, Min Kim

**Affiliations:** 1Department of Electronic Engineering, Pukyong National University, 45 Yongso-ro, Nam-gu, Busan 48513, Republic of Korea; alex987@naver.com (S.Y.); radar@pknu.ac.kr (S.P.); 2Department of Maritime ICT & Mobility Research, Korea Institute of Ocean Science & Technology, 385, Haeyang-ro, Yeongdo-gu, Busan 49111, Republic of Korea; baeksj@kiost.ac.kr; 3Department of Smart Mobility Engineering, Pukyong National University, 45 Yongso-ro, Nam-gu, Busan 48513, Republic of Korea; inoh@pknu.ac.kr; 4Radsys, 22, 12 Maegok-ro, Dasa-eup, Dalseong-gun, Daegu 42908, Republic of Korea; ksbget@radsys.kr; 5Intelligent Semiconductor Research Division, Electronics and Telecommunications Research Institute, 218 Gajeong-ro, Yuseong-gu, Daejeon 34129, Republic of Korea; koobt@etri.re.kr (B.K.); ysbaek@etri.re.kr (Y.B.); 6The Institute of Security Convergence Technology, Konkuk University, 268, Chungwon-daero, Chungju-si 27478, Republic of Korea; jungjooho68@hanmail.net

**Keywords:** vital sign estimation, motion compensation, multiple signal classification, short-range radar, principal component analysis

## Abstract

Various short-range radars, such as impulse-radio ultra-wideband (IR-UWB) and frequency-modulated continuous-wave (FMCW) radars, are currently employed to monitor vital signs, including respiratory and cardiac rates (RRs and CRs). However, these methods do not consider the motion of an individual, which can distort the phase of the reflected signal, leading to inaccurate estimation of RR and CR because of a smeared spectrum. Therefore, motion compensation (MOCOM) is crucial for accurately estimating these vital rates. This paper proposes an efficient method incorporating MOCOM to estimate RR and CR with super-resolution accuracy. The proposed method effectively models the radar signal phase and compensates for motion. Additionally, applying the super-resolution technique to RR and CR separately further increases the estimation accuracy. Experimental results from the IR-UWB and FMCW radars demonstrate that the proposed method successfully estimates RRs and CRs even in the presence of body movement.

## 1. Introduction

Non-contact vital sign estimation (VSE) techniques using radar systems have been extensively researched and employed for various applications [1,2,3,4,5,6,7,8,9,10,11,12,13,14]. These techniques estimate vital signs, such as respiratory and cardiac rates (RRs and CRs), by detecting very small range variations in the skin caused by these vital signs. Because lung and heart vibrations contribute to the movement of the chest and back, the phase of the reflected radar signal contains RR and CR information [15]. Numerous types of short-range radars have been used to accurately extract these phases for VSE [1,2,3,6].

To achieve accurate and cost-effective VSE, various studies have explored the use of continuous-wave (CW), frequency-modulated continuous-wave (FMCW), and impulse-radio ultra-wideband (IR-UWB) signals. One study proposed an efficient CW radar system for RR measurement under one-dimensional body motion [16]. Another method introduced motion compensation (MOCOM) to eliminate phase errors caused by random body movements [17]. This approach assumes a CW signal and uses two echo signals from both the chest and back to cancel phase errors. Schires [18] developed an IR-UWB radar system integrated into a car seat backrest, matched to the human body. Xiang [19] presented a high-precision VSE method using FMCW radar, in which noise was considerably reduced via a variation-mode decomposition (VMD) wavelet-internal-thresholding algorithm. A recently proposed method used a fuzzy rule to accurately determine RR and remove distorted CR values due to motion [20].

Additionally, substantial research has been directed at improving both hardware and software techniques for VSE. Liu et al. [21] suggested an interferometry-based frequency-sweeping technique to achieve high-ranging accuracy and a wide detection range. Gu [22], and Yavari [23] introduced methods based on digital post-distortion and compensation to enhance accuracy by mitigating undesirable interference. To further increase accuracy and suppress noise, signal decomposition methods such as wavelet transform [24] and empirical mode decomposition (EMD) [25] have been applied to VSE.

However, existing methods do not fully address the issues caused by considerable movement of an individual. Continuous motion during observation can result in substantial phase errors. The method described in [16] may fail because, generally, the movement is not one-dimensional. The approach in [17] assumes that the phases caused by forward and backward motions have opposite signs and equal magnitudes, allowing error removal by multiplying the spectra of the forward and backward radars. This method can fail if other motion components are present. Additionally, because the amplitude of the signal reflected from the front of the body is significantly larger than that from the back, spectrum multiplication cannot eliminate forward–backward motion artifacts.

The results in [18] may be significantly degraded if the back moves or if the matching condition between the body and the seat changes. The method in [19] did not incorporate MOCOM, and thus, it could fail with body movement. The approach in [20] fails if the estimated value remains inaccurate over time, as it maintains the previous result once distortion is detected. Methods in [21,22,23] are not relevant to MOCOM and may be inaccurate in the presence of movement. While decomposition methods [24,25] can successfully separate signals in low-noise environments, their accuracy is questionable owing to spectrum blurring, even at very low velocities and accelerations. Moreover, when respiratory and cardiac activities are partially blocked or the radar line-of-sight changes rapidly, the signal-to-noise ratio (SNR) can decrease significantly, leading to poor separation results. Therefore, an efficient method to eliminate the effects of motion is required.

In this study, we propose an efficient method that achieves super-resolution vital sign estimation (VSE) using a low-cost short-range radar. The proposed method involves the following steps:Denoising and signal separation: the radar signal is denoised using principal component analysis (PCA).Motion compensation (MOCOM): based on an analysis of the phase components of the echo signal, two efficient MOCOM methods, MOCOM #1 and MOCOM #2, are introduced.Noise reduction and auto-focusing: to further reduce noise and enhance resolution, the respiratory and cardiac signals are separated again and auto-focused.Super-resolution spectrum estimation: the multiple signal classification (MUSIC) method [26], a super-resolution technique, is used to obtain the spectra of the separated signals with very high resolution.

The remainder of this paper is organized as follows: Section 2 introduces the radar signal model, analyzing the phase components and issues caused by movement. Section 3 details the proposed method. Section 4 presents experimental results demonstrating the method’s efficiency, and Section 5 concludes the paper.

## 2. Signal Model and Problem Analysis

### 2.1. Radar Signal Model

To apply the proposed method to various low-cost radars, the signal models of two representative short-range radars were assumed to be IR-UWB and FMCW. Using the observation geometry shown in Figure 1, the methods employed in previous studies [20,27] were modified, where $r_{RLOS}={\left[1,0,0\right]}^{T}$ is the radar line-of-sight (RLOS) vector. When there is no body movement, the range vector $r_{r}\left(t\right)$ of the chest caused by RR is $r_{r}\left(t\right)={\left[d_{r}\cos\left(2\pi f_{r}\left(t\right)t+\varphi_{r0}\right),0,0\right]}^{T}$ and the range variation along the RLOS is given by
(1)$$R_{r}\left(t\right)=r_{RLOS}^{T}\cdot r_{r}\left(t\right)=d_{r}\cos\left(2\pi f_{r}\left(t\right)t+\varphi_{r0}\right),$$
where · denotes the inner product operator, $d_{r}$ denotes the variation in chest skin, $\varphi_{r0}$ represents the initial phase, and
(2)$$f_{r}\left(t\right)=f_{r0}+▵f_{r}\cos\left(2\pi f_{r0}t\right),$$
is the real-time respiration rate, where $f_{r0}$ denotes the fundamental RR frequency and $▵f_{r}$ represents the slight variation amplitude in the respiration rate responsible for high-order harmonics.

Similarly, the range vector $r_{c}\left(t\right)$ of the chest caused by the CR is $r_{c}\left(t\right)=[d_{c}\cos(2\pi f_{c}\left(t\right)t$$+\varphi_{c0}{),0,0]}^{T}$ and its projection $R_{c}\left(t\right)$ along the RLOS is
(3)$$R_{c}\left(t\right)=r_{RLOS}^{T}\cdot r_{c}\left(t\right)=d_{c}\cos\left(2\pi f_{c}\left(t\right)t+\varphi_{c0}\right),$$
where $d_{c}$ is the variation in the chest skin, $\varphi_{c0}$ is the initial phase, and
(4)$$f_{c}\left(t\right)=f_{c0}+▵f_{c}\cos\left(2\pi f_{c0}t\right),$$
is the real-time CR function.

For the transmitted signal, we assumed the following functions of the IR-UWB and FMCW radars with bandwidth *B*:(5)$$s_{t}\left(t\right)=\sqrt{P_{t}}f_{IR}\left(t\right)\;\mathrm{and}\;s_{t}\left(t\right)=\sqrt{P_{t}}f_{FMCW}\left(t\right),$$
where
(6)$$f_{IR}\left(t\right)=gauss\left(Bt\right)\exp\left(-j2\pi f_{cen}t\right),$$
(7)$$f_{FMCW}\left(t\right)=\exp\left(j2\pi \left(f_{cen}t+\frac{K_{r}t^{2}}{2}\right)\right),$$
where $\sqrt{P_{t}}$ denotes the amplitude, gauss(Bt) denotes the Gaussian pulse with *B*, $f_{cen}$ denotes the carrier frequency of the transmitted signal, $K_{r}=B/T_{chirp}$, and $T_{chirp}$ denotes the width of the transmitted chirp. The received FMCW signal is compressed into a sinc function by de-chirping and employing a Fourier transform (FT) [28].

For the chest observed with a given pulse repetition time $T_{PRI}$, the received signal of the IR-UWB and FMCW radars is expressed as
(8)$$s_{r}\left(t,t_{p}\right)=\sqrt{P_{r}}psf\left(B(t-\tau \left(t_{p}\right))\right)\exp\left(j2\pi f_{cen}\left(t-\tau \left(t_{p}\right)\right)\right),$$
where psf(Bt)=gauss(Bt) and sinc(Bt) for the IR-UWB and FMCW radars, respectively. $t_{p}$ is the slow time sampled at $T_{PRI}$, and the time delay $\tau (t_{p})$ is
(9)$$\tau \left(t_{p}\right)=\frac{2}{c}R_{w}\left(t_{p}\right)+\frac{2}{c}R_{r}\left(t_{p}\right)+\frac{2}{c}R_{c}\left(t_{p}\right).$$
*c* is the velocity of light, and $R_{w}\left(t_{p}\right)$ is the range variation of the body that should be removed. $\sqrt{P_{r}}$ is the amplitude of the received signal and can be expressed by the radar equation as follows:(10)$$P_{r}=\frac{P_{t}G_{t}G_{r}\lambda^{2}\sigma_{RCS}}{{\left(4\pi \right)}^{3}R^{4}},$$
where $G_{t}$ and $G_{r}$ are the gains of the transmitting and receiving antennas, respectively; λ is the wavelength, $\sigma_{RCS}$ is the radar cross-section, and *R* is the distance to the chest.

$s_{r}\left(t,t_{p}\right)$ is down-converted to baseband signal $s_{B}\left(t,t_{p}\right)$ as follows:(11)$$\begin{array}{cc} s_{B}\left(t,t_{p}\right) & =s_{r}\left(t,t_{p}\right)\exp\left(-j2\pi f_{cen}t\right)\\ \; & =\sqrt{P_{r}}psf\left(B(t-\tau \left(t_{p}\right))\right)\exp\left(-j2\pi f_{cen}\tau \left(t_{p}\right)\right). \end{array}$$

Assuming $R_{w}\left(t_{p}\right)=R_{0}-v_{0}t_{p}-0.5a_{0}t_{p}^{2}$, where $R_{0}$ is the initial range and $v_{0}$ and $a_{0}$ are the velocity and acceleration of the body, respectively, (Equation 11) becomes
(12)$$\begin{array}{cc} s_{B}\left(t,t_{p}\right) & =\sqrt{P_{r}}psf\left(B(t-\tau \left(t_{p}\right))\right)\exp\left(-j2\pi f_{cen}\frac{2}{c}R_{0}\right)\\ \; & \times \exp\left(j2\pi f_{cen}\frac{2}{c}\left(v_{0}t_{p}+\frac{1}{2}a_{0}t_{p}^{2}\right)\right)\\ \; & \times \exp\left(-j2\pi f_{cen}\frac{2}{c}R_{r}\left(t_{p}\right)\right)\exp\left(-j2\pi f_{cen}\frac{2}{c}R_{0}\right). \end{array}$$

### 2.2. Effect of the Motion of the Rigid Body

The clipped signal $s_{cp}\left(t_{p}\right)$ at range *r* with maximum energy is a function of $t_{p}$ only and contains information on $f_{r}$ and $f_{c}$. $s_{cp}\left(t_{p}\right)$ is given by
(13)$$\begin{array}{cc} s_{cp}\left(t_{p}\right)= & \sqrt{P_{r}}\exp\left(-j2\pi f_{cen}\frac{2}{c}R_{0}\right)\\ & \times \exp\left(j2\pi f_{cen}\frac{2}{c}\left(v_{0}t_{p}+\frac{1}{2}a_{0}t_{p^{2}}\right)\right)\\ & \times \exp\left(-j2\pi f_{cen}\frac{2}{c}R_{r}\left(t_{p}\right)\right)\\ & \times \exp\left(-j2\pi f_{cen}\frac{2}{c}R_{c}\left(t_{p}\right)\right). \end{array}$$

The Fourier transform of $s_{cp}\left(t_{p}\right)$ can be expressed as
(14)$$\begin{array}{cc} s_{cp}\left(t_{p}\right)= & \sqrt{P_{r}}\exp\left(-j2\pi f_{cen}\frac{2}{c}R_{0}\right)\\ & ⊗FT\left\{\exp\left(j2\pi f_{cen}\frac{2}{c}\left(v_{0}t_{p}+\frac{1}{2}a_{0}t_{p^{2}}\right)\right)\right\}\\ & ⊗FT\left\{\exp\left(-j2\pi f_{cen}\frac{2}{c}R_{r}\left(t_{p}\right)\right)\right\}\\ & ⊗FT\left\{\exp\left(-j2\pi f_{cen}\frac{2}{c}R_{c}\left(t_{p}\right)\right)\right\}. \end{array}$$
where FT{·} denotes the FT operation, and ⊗ denotes the convolution operation.

As the first component is a constant envelope of one, its effect can be eliminated. The FT of the second exponential component in the above equation is a linear frequency modulation signal due to the acceleration shifted by the Doppler frequency $f_{0}=2f_{c}v/c$. Using the stationary phase method [29], it can be transformed into
(15)$$G\left(f\right)=Rect\left(\frac{f-f_{0}-KT_{p}/2}{KT_{p}}\right)\exp\left(\frac{-j\pi {\left(f-f_{0}\right)}^{2}}{K}\right),$$
where Rect(·) is a rectangle function, $T_{p}$ is the coherent processing interval, and *K* is given by
(16)$$K=-2f_{c}a_{0}/c.$$

The FT of the third and fourth components can be divided into in-phase and quadrature components as follows:(17)$$FT\left\{\exp(j4\pi f_{cen}R_{r}\left(t_{p}\right)/c\right\}=I_{r}\left(t_{p}\right)+Q_{r}\left(t_{p}\right),$$
(18)$$FT\left\{\exp(j4\pi f_{cen}R_{c}\left(t_{p}\right)/c\right\}=I_{c}\left(t_{p}\right)+Q_{c}\left(t_{p}\right),$$
where $I_{r}\left(t_{p}\right)$ and $Q_{r}\left(t_{p}\right)$ are the in-phase and quadrature components of RR, respectively, and $I_{c}\left(t_{p}\right)$ and $Q_{c}\left(t_{p}\right)$ are those of CR. $I_{r}\left(t_{p}\right)$, $Q_{r}\left(t_{p}\right)$, $I_{c}\left(t_{p}\right)$, and $Q_{c}\left(t_{p}\right)$ can be approximated using the Bessel function $J_{i}\left(x\right)$ of the first type of order *i* as follows [17]:(19)$$I_{r}\left(t_{p}\right)=2\left(C_{20}\cos\left(2\omega_{r}t_{p}\right)+C_{02}\cos\left(2\omega_{r}t_{p}\right)+\ldots \right),$$
(20)$$Q_{r}\left(t_{p}\right)=2\left(C_{10}\sin\left(\omega_{r}t_{p}\right)+C_{01}\sin\left(\omega_{r}t_{p}\right)+\ldots \right),$$
(21)$$I_{c}\left(t_{p}\right)=2\left(C_{20}\cos\left(2\omega_{c}t_{p}\right)+C_{02}\cos\left(2\omega_{c}t_{p}\right)+\ldots \right),$$
(22)$$Q_{r}\left(t_{p}\right)=2\left(C_{10}\sin\left(\omega_{c}t_{p}\right)+C_{01}\sin\left(\omega_{c}t_{p}\right)+\ldots \right),$$
where $\omega_{r}=2\pi f_{r}$, $\omega_{c}=2\pi f_{c}$, and $C_{ij}=J_{i}\left(4\pi f_{cen}d_{r}/c\right)J_{j}\left(4\pi f_{cen}d_{c}/c\right)$ for i=0,1,…,∞ and j=0,1,…,∞. Notably, *i* and *j* cannot be zero simultaneously, as they represent the DC component.

Equations (Equation 2), (Equation 4), and (Equation 19)–(Equation 22) represent spectra of RR and CR, composed of harmonics of $f_{r}$ and $f_{c}$; as $J_{i}\left(x\right)$ is higher for smaller *i* and RR has a larger $d_{r}$ (e.g., $0.001\leq d_{r}\leq 0.003$ m) than $d_{c}$ (e.g., $0.00005\leq d_{c}\leq 0.0005$ m), $f_{r}$ has the largest amplitude. However, the harmonics of $f_{r}$ may have negative effects on $f_{c}$ when they coincide with $f_{c}$.

The convolution of the discrete spectra of (Equation 17) and (Equation 18) with the FT of (Equation 15) yielded a distorted spectrum owing to the frequency modulation caused by the acceleration and shift of the modulated spectrum (Figure 1). In the case where $v_{0}\neq 0\;$ m/s and $a_{0}=0\;$ ${m/s}^{2}$, a peak appears at a false location owing to $f_{0}$, and when $v_{0}=0\;$ m/s and $a_{0}\neq 0\;$ ${m/s}^{2}$, the spectrum can be blurred because of the convolution with the Doppler bandwidth caused by $a_{0}$. With nonzero $v_{0}$ and $a_{0}$, the spectrum is severely distorted, as the peaks are both displaced and blurred. Therefore, MOCOM is an indispensable preprocessing step for accurate RRE and CRE.

Eliminating the effect of $v_{0}$ and $a_{0}$ using MOCOM is challenging. One probable method is to estimate $v_{0}$ first by identifying the largest peak in $S_{cp}\left(f\right)$ and subsequently focusing the spectrum by finding $a_{0}$. However, the exact value of $v_{0}$ can seldom be found because the shape of the spectrum caused by $a_{0}$ may not be flat; because of Gibbs’ phenomenon, overshot values can be larger than that at $f_{0}$, and the fluctuation of the Doppler bandwidth owing to $a_{0}$ and the interference by neighboring harmonics makes the spectrum more irregular. Therefore, $S_{cp}\left(f\right)$ should not occur in the presence of $v_{0}$ and $a_{0}$.

The problem of finding peaks in the spectrum of $S_{cp}\left(f\right)$ can be solved simply by using the phase information, that is, the arctangent demodulation $\theta (t_{p})$ of $S_{cp}\left(t\right)$, given as
(23)$$\begin{array}{cc} \theta (t_{p}) & =\arctan\left\{Q_{cp}\left(t_{p}\right)/I_{cp}\left(t_{p}\right)\right\}\\ \; & \approx \left\{4\pi /\lambda_{c}\right\}\left\{R_{0}+R_{w}\left(t_{p}\right)+R_{r}\left(t_{p}\right)+R_{c}\left(t_{p}\right)\right\}\\ \; & \approx \theta_{0}+\theta_{w}\left(t_{p}\right)+\theta_{r}\left(t_{p}\right)+\theta_{c}\left(t_{p}\right), \end{array}$$
where $I_{cp}\left(t_{p}\right)$ and $Q_{cp}\left(t_{p}\right)$ are the in-phase and quadrature components of $s_{cp}\left(t_{p}\right)$, respectively and $\lambda_{c}$ is the wavelength of $f_{cen}$. $\theta (t_{p})$ is significantly distorted by $\theta_{w}\left(t_{p}\right)$ because it is considerably larger than $\theta_{r}\left(t_{p}\right)$ and $\theta_{c}\left(t_{p}\right)$ (Figure 2). To avoid ambiguity in extracting $\theta (t_{p})$, a phase unwrapping technique should be used [20].

Considering the effects of the surrounding clutter and noise, (Equation 23) is modified as follows:(24)$$\theta \left(t_{p}\right)\approx \theta_{0}+\theta_{w}\left(t_{p}\right)+\theta_{r}\left(t_{p}\right)+\theta_{c}\left(t_{p}\right)+\theta_{noise}\left(t_{p}\right)+\theta_{clutter}\left(t_{p}\right),$$
where $\theta_{noise}\left(t\right)$ and $\theta_{clutter}\left(t\right)$ are the phases of system noise and background clutter, respectively. As $\theta (t_{p})$ includes a linear mixture of the effects of the body, noise, and clutter, RRE and CRE become more challenging.

When Fourier transformed, $FT\theta (t_{p})$ is the linear component of each component, as follows:
(25)$$\begin{array}{cc} FT\{\theta \left(t_{p}\right)\}= & FT\left\{\theta_{0}\right\}+FT\left\{\theta_{w}\left(t_{p}\right)\right\}+FT\left\{\theta_{r}\left(t_{p}\right)\right\}\\ & +FT\left\{\theta_{c}\left(t_{p}\right)\right\}+FT\left\{\theta_{noise}\left(t_{p}\right)\right\}\\ & +FT\{\theta_{clutter}\left(t_{p}\right)\}, \end{array}$$
Because $\theta_{0}=4\pi /\lambda_{c}/R_{0}$ is constant, $FT\theta (t_{p})$ has a large amplitude at f=0 Hz. In addition, $\theta_{r}\left(t\right)$ and $\theta_{c}\left(t\right)$ are the two cosine functions of $f_{r}$ and $f_{c}$, respectively, and the peaks can appear at the corresponding frequencies. However, the effect of $\theta_{w}\left(t_{p}\right)$ should be removed before the FT of $\theta (t_{p})$, because it has a very large amplitude (see Figure 3). In addition, the effects of $\theta_{noise}\left(t_{p}\right)$ and $\theta_{clutter}\left(t_{p}\right)$ should be minimized.

## 3. Proposed Method

### 3.1. Summary of the Proposed Method

This subsection introduces an efficient method for estimating the vital parameters $f_{r}$ and $f_{c}$ with super-resolution by compensating for motion and applying the MUSIC algorithm (Figure 4).

①Range alignment: the range profile history is aligned to position the scatterer at the same location during the coherent processing interval, thereby eliminating range migration caused by movement.②Radar signal clipping: the radar signal is clipped around the range bin with the maximum energy to enhance SNR.③PCA denoising: PCA helps isolate the vital signals from the background noise and clutter, which is critical for accurate estimation.④Removal of the rigid-body Doppler: the Doppler frequency corresponding to the rigid body’s velocity is estimated using a fast Fourier transform (FFT) and is then compensated.⑤Phase unwrapping: a phase unwrapping technique is applied to extract the phase history.⑥MOCOM ♯1 (Gaussian filtering): The phase history is coarsely estimated using a Gaussian filter to remove phase errors caused by body movement.⑦MOCOM ♯2 (Envelope fitting): the envelope of the phase history is fitted to minimize residual errors that were not compensated for in MOCOM ♯1.⑧Signal reconstruction: the complex signal is reconstructed using amplitude- and motion-compensated phases.⑨Signal separation: low-pass (LPF) and high-pass filters (HPF) are used to separate the respiratory and cardiac signals.⑩Phase adjustment: residual phase errors are removed using a phase-adjustment technique for each of the separated signals.⑪Vital parameters estimation: $f_{r}$ and $f_{c}$ are estimated by identifying the maximum peaks in each super-resolution spectrum, which are generated via zero-padding and the MUSIC algorithm.

### 3.2. Main Idea of the Proposed Method

#### 3.2.1. Range Alignment

First, because $\tau_{p}\left(t_{p}\right)$ in (Equation 7) is heavily dependent on $v_{0}$ and $a_{0}$, the scatterer migrates into neighboring range bins. Consequently, $s_{cp}\left(t_{p}\right)$ cannot be formed by clipping $s_{B}\left(r,t_{p}\right)$ at a fixed range r=ct/2. Therefore, the RPs should be aligned such that the scatterer is located in the same range bin for all $t_{p}$. While exact estimation of motion parameters $v_{0}$ and $a_{0}$ is impossible without errors, we used a widely adopted non-parametric method. Among various methods, we employed entropy minimization, which is commonly used in inverse synthetic aperture radar imaging [30].

The entropy between two RPs ${\mathrm{rp}}_{1}\left(r\right)$ and ${\mathrm{rp}}_{2}\left(r-\varepsilon \right)$ is defined by
(26)$$H_{rp_{1},rp_{2}}\left(\varepsilon \right)=-\int_{0}^{r_{max}}{\mathrm{rp}}_{av}\left(r\right)\ln{\mathrm{rp}}_{av}\left(r\right)dr,$$
where
(27)$${\mathrm{rp}}_{av}\left(r\right)=\frac{|{\mathrm{rp}}_{1}\left(r\right)|+|{\mathrm{rp}}_{2}\left(r-\varepsilon \right)|}{\int_{0}^{r_{max}}\left(\right|{\mathrm{rp}}_{1}\left(r\right)|+|{\mathrm{rp}}_{2}\left(r-\varepsilon \right)\left|\right)dr},$$
and $r_{max}=T_{PRI}\cdot c/2$ is the maximum range. According to this criterion, the ε value that minimizes the 1D entropy is the shift that aligns well with ${\mathrm{rp}}_{2}\left(t\right)$. Consequently, the second RP of $s_{B}\left(r,t_{p}\right)$ and the relative shift ε minimizing (Equation 26) along with the average of the aligned RPs are determined. The RP is aligned with a shift by ε.

#### 3.2.2. Extraction of Vital Signals and Denoising Using PCA

Because the RPs of the body can be contaminated by noise or surrounding clutter, the phase of the RP can be distorted. In addition, determining the exact location of the chest is difficult, because the reflected signal is the sum of several body parts, mainly the torso, which can be buried within the torso signal. Therefore, we applied PCA to denoise and decompose vital signals.

Assuming *n* RPs, let A be an m×n matrix constructed by sampling ±m/2 range bins around the range bin with the maximum energy as follows:(28)$$A={\left[o_{0},o_{1},...,o_{m-1}\right]}^{T},$$
where
(29)$$o_{k}={\left[{\mathrm{rp}}_{k}\left(r_{0}\right),{\mathrm{rp}}_{k}\left(r_{1}\right),...,{\mathrm{rp}}_{k}\left(r_{n-1}\right)\right]}^{T},$$
the covariance matrix is calculated by
(30)$$\mathrm{Cov}=\frac{1}{m-1}\sum_{k=0}^{m-1}o_{k}o_{k}^{H},$$
where *H* is the conjugate transpose of the vector. Subsequently, the eigenvalue decomposition of Cov is used, given by
(31)$$\mathrm{Cov}=E\Lambda E^{T}.$$

According to PCA, the orthonormal eigenvectors corresponding to large eigenvalues span the signal subspace of the Cov, while those corresponding to small eigenvalues span the noise subspace. Additionally, because the large-valued torso signal corresponds to A projected onto the eigenvector $e_{1}$ with the largest eigenvalue, the small-valued respiratory and cardiac signals can be separated by projections onto eigenvectors with smaller eigenvalues. The matrix A is projected onto $E_{prj}=\left[e2e3\right]$ as
(32)$$A_{prj}=\left[a_{1},a_{2}\right]={\mathrm{AE}}_{prj},$$
Finally, the denoised sum of the respiratory and the cardiac signal is
(33)$$s_{sum}=a_{1}+a_{2},$$
and it is used instead of the $s_{cp}\left(t_{p}\right)$ in (Equation 13). Note that $s_{sum}$ is a function of $t_{p}$.

#### 3.2.3. Estimation and Removal of the Rigid-Body Doppler

The Doppler frequency $f_{D}$ corresponding to the rigid body velocity $v_{0}$ is estimated by
(34)$${\hat{f}}_{D}=\underset{f}{\mathrm{argmax}}\left\{|FFT\{s_{sum}\}|\right\},$$
where FFT· denotes the FFT function. The rigid-body phase error is removed using
(35)$$s_{cp}\left(t_{p}\right)=s_{sum}\left(t_{p}\right)\exp\left(j2\pi {\hat{f}}_{D}t_{p}\right).$$

#### 3.2.4. MOCOM #1

Although the rigid-body Doppler is removed using (Equation 35), the spectrum can be blurred by the effect of the acceleration $a_{0}$ of the rigid body. In addition, because the FT of the acceleration component yields a Doppler bandwidth with an irregular passband (Figure 1), the estimation of v0 may not be accurate. Consequently, Equation (Equation 35) may yield an estimation error, resulting in a shift in the frequency components of $S_{cp}\left(f\right)$.

This problem can be solved using only unwrapped $\theta (t_{p})$. $\theta (t_{p})$ includes the linear sum of $\theta_{w}\left(t_{p}\right)$, $\theta_{r}\left(t_{p}\right)$, and $\theta_{c}\left(t_{p}\right)$, and $\left(v_{0},a_{0}\right)$ can be addressed separately. Because the spectrum is severely blurred in the frequency domain of the phase, extracting this component by applying signal-separation methods is difficult. In addition, because the shape of the unwrapped phase includes the effects of motion, removing the motion component from the time domain is more profitable.

The error can be perfectly removed once the motion component is accurately estimated using a proper polynomial, such as $P\left(t_{p}\right)=a_{0}+a_{1}t_{p}^{1}+a_{2}t_{p}^{2}+...+a_{q}t_{p}^{q}$ to describe $\theta (t_{p})$, and a curve-fitting algorithm to find the parameter $P(t_{p})$. However, the curve-fitting algorithm is time-consuming, and numerous phase errors can occur if $P(t_{p})$ does not match $\theta (t_{p})$ accurately. Therefore, we coarsely compensated for the motion using a smoothing kernel function to smooth $\theta (t_{p})$. Among the various methods, we used the following Gaussian kernel:(36)$$K\left(t_{p}\right)=\frac{dt_{p}}{\sqrt{2\pi }\sigma }\exp\left(-\frac{{\left(t_{p}-\mu_{t_{p}}\right)}^{2}}{2\sigma^{2}}\right),$$
where $dt_{p}$ is the sampling interval of $t_{p}$, σ is the standard deviation of the kernel, and $\mu_{t_{p}}$ is the mean value of $t_{p}$.

The smoothed phase for the MOCOM is given by
(37)$$\theta_{s}\left(t_{p}\right)=\theta \left(t_{p}\right)⊗K\left(t_{p}\right).$$

Equation (Equation 37) is time consuming; therefore, it is converted in the frequency domain as follows:(38)$$\theta_{s}\left(t_{p}\right)=IFT\left\{FT\left\{\theta \left(t_{p}\right)\right\}\times FT\left\{K\left(t_{p}\right)\right\}\right\},$$
where IFT· is the inverse FT. Finally, by directly subtracting $\theta_{s}\left(t_{p}\right)$ from $\theta (t_{p})$ as follows:(39)$$\theta_{M1}\left(t_{p}\right)=\theta \left(t_{p}\right)-\theta_{s}\left(t_{p}\right),$$
the effect of $\theta_{w}\left(t_{p}\right)$ is eliminated. This method is direct and relatively simple (Figure 5). However, $\theta_{M1}\left(t_{p}\right)$ includes errors because various errors caused by phase unwrapping and noise cause a mismatch between $\theta_{s}\left(t_{p}\right)$ and the actual phase trajectory. In Figure 5, red and blue lines represent $\theta_{M1}\left(t_{p}\right)$ and the residual error caused by this mismatch, respectively. As a result, additional MOCOM is required. Therefore, we define this MOCOM as MOCOM #1 to distinguish it from the additional MOCOM, MOCOM #2.

#### 3.2.5. MOCOM #2

The MOCOM #2 proposed in this study reduces the residual error, as shown in Figure 5. Owing to the sinusoidal nature of the respiratory and cardiac signals, the sum of the motion-compensated signals in the time domain fluctuates within constant upper and lower limits (Figure 6). Therefore, the ideal MOCOM flattens the phase by estimating the unknown residual error, as shown in Figure 3. A parametric method is ineffective for estimating that curve. Instead, we successively use three phase envelopes after MOCOM #1: the upper, lower, and mean envelopes (Figure 7).

Various methods for detecting the envelope of a signal in a communication area exist, such as squaring, low-pass filtering, and the Hilbert transform [31,32]. In this study, we use a simple detection method by using the maximum (or minimum) within a window with length *L*. By moving a window from left to right with a given time step $▵t_{w}$, the maximum value within the window is stored as $\theta_{max}\left(k▵t_{w}\right)$, and the minimum value is stored as $\theta_{min}\left(k▵t_{w}\right)$, where $k=0,1,2,...,round(T_{f}/▵t_{w})$. $T_{f}$ is the length of the time frame, and round() is the function that determines the nearest integer (Figure 8). Zeros are added when the number of signals is smaller than *L* near the start and end of $\theta_{M1}\left(t_{p}\right)$.

The upper envelope $E_{upper}\left(t_{p}\right)$ and lower envelope $E_{lower}\left(t_{p}\right)$ are then obtained by curve-fitting $\theta_{max}\left(k▵t_{w}\right)$ and $\theta_{min}\left(k▵t_{w}\right)$. Among various methods, we use the cubic spline interpolation method [33]. Finally, the mean envelope is obtained as
(40)$$E_{mean}\left(t_{p}\right)=\left(E_{upper}\left(t_{p}\right)+E_{lower}\left(t_{p}\right)\right)/2,$$
and the error is reduced by
(41)$$\theta_{M2}\left(t_{p}\right)=\theta_{M1}\left(t_{p}\right)-E_{mean}\left(t_{p}\right).$$

However, depending on the length of the window *L* and time step $▵t_{w}$, the envelope may not describe the overall shape of $\theta_{M1}\left(t_{p}\right)$. In particular, near $t_{p}=0$ and $t_{p}=T_{p}$, the estimated envelope may increase or decrease significantly, owing to the zeros added to the window. In addition, the maximum and minimum values at a certain time may be the same, yielding a large difference from the near maximum or minimum value. In this case, $\theta_{M2}\left(t_{p}\right)$ may not be flat but curved near that point. This problem can be solved by applying MOCOM #2 successively until the variance of $E_{mean}\left(t_{p}\right)$ is close to 0; that is,
(42)$$var(E_{mean}\left(t_{p}\right))\approx 0.$$

#### 3.2.6. Reconstruction of Signals and Separation of Vital Signals Using LPF and HPF

To further remove the residual phase errors and obtain super-resolution for $f_{r}$ and $f_{c}$, the complex signal is first reconstructed as follows:(43)$$s_{M2}\left(t_{p}\right)=\left|s_{cp}\left(t_{p}\right)\right|\exp\left(j\theta_{M2}\left(t_{2}\right)\right).$$
Subsequently, (Equation 43) is separated into respiratory and cardiac signals, that is, $s_{R}\left(t_{p}\right)$ and $s_{C}\left(t_{p}\right)$, respectively. Because the amplitudes of the cardiac spectrum are very small and can be influenced by the ripples of the passband, we used a maximally flat Butterworth filter: LPF for the respiratory signal and HPF for the cardiac signal, in the following form:(44)$$HPF\left(\omega \right)=\frac{b_{1}+b_{2}e^{-j\omega }+b_{3}e^{-j2\omega }+...+b_{N_{B}}e^{-j(N_{B}-1)\omega }}{a_{1}+a_{2}e^{-j\omega }+a_{3}e^{-j2\omega }+...+a_{N_{B}}e^{-j(N_{B}-1)\omega }},$$
where $N_{B}$ is the filter order.

#### 3.2.7. Phase Adjustment

Despite its maximum flatness, $\theta_{M2}\left(t_{p}\right)$ contains only CR and RR frequencies. The residual phase errors may occur owing to estimation error, noise, and clutter. Consequently, residual errors should be removed using an appropriate method. The residual error is significantly reduced by using MOCOM #1 and #2; therefore, we used the 1D entropy of the following signal spectrum:(45)$$s_{fil}\left(t_{p}\right)=\left|s_{fil}\left(t_{p}\right)\right|\exp\left(j\theta_{fil}\left(t_{p}\right)\right),$$
where $s_{fil}\left(t_{p}\right)$ is the filtered signal, representing either $s_{R}\left(t_{p}\right)$ and $s_{C}\left(t_{p}\right)$. Subsequently, the phase-error $\hat{\theta }\left(t_{p}\right)$ is determined by minimizing the entropy, defined by
(46)$$H\left(s_{PA}\left(t_{p}\right)\right)=-\int_{-\infty }^{\infty }\left|S_{PA}\left(f\right)\right|\ln|S_{PA}\left(f\right)|df,$$
where
(47)$$s_{PA}\left(t_{p}\right)=s_{fil}\left(t_{p}\right)\exp\left(\hat{\theta }\left(t_{p}\right)\right),$$
and $S_{PA}\left(f\right)=FT\left\{s_{PA}\left(t_{p}\right)\right\}$. This method is a major topic of interest and is known as phase adjustment in radar imaging. In this study, we applied the widely used minimum entropy phase adjustment (MEPA) [34]. The MEPA is fast and can be conducted in real-time using the entropy gradient.

#### 3.2.8. Zero-Padding and Application of MUSIC

To improve estimation accuracy, the phase-adjusted signal is zero-padded to upsample the spectrum. Subsequently, MUSIC is applied to the zero-padded (optional) signal to obtain a super-resolution spectrum. MUSIC uses the characteristic that the direction vector containing the signal is orthogonal to the noise subspace consisting of noise eigenvectors.

Using the vector y containing *N* samples of the phase $\theta_{PA}\left(t_{p}\right)$ of the radar signal after the phase adjustment, which is given by
(48)$$\begin{array}{cc} y & =[y_{0},y_{1},...,y_{N-1}]\\ \; & =[\theta_{PA}\left(0\right),\theta_{PA}\left(T_{PRI}\right),...,\theta_{PA}\left(\left(N-1\right)T_{PRI}\right)], \end{array}$$
the covariance matrix $R_{yy}=E\left[{\mathrm{yy}}^{H}\right]$, where E[·] is the ensemble average, is estimated by applying the modified spatial smoothing preprocessing to decorrelate signals from various frequency components as follows:(49)$$R_{yy}=\frac{1}{2M}\sum_{k=0}^{M_{sub}-1}\left(R_{k}+{\mathrm{JR}}_{k}^{*}J\right),$$
where
(50)$$R_{k}=y_{k}y_{k}^{H},$$
(51)$$y_{k}={\left[y_{k},y_{k+1},...,y_{k+N_{sub}}-1\right]}^{T},$$
and
(52)$$J=\left[\begin{array}{cccc} 0 & 0 & ... & 1\\ 0 & ... & 1 & 0\\ \vdots & \ddots & \vdots & \vdots \\ 1 & 0 & ... & 0 \end{array}\right].$$
In (Equation 49), $M_{sub}$ is the number of subarrays, and $N_{sub}$ is the subarray dimension. Equation (Equation 47) is the average $R_{k}$ obtained by $y_{k}$ clipped by two windows of size $N_{sub}$ moving forward and backward [26].

Assuming the number of frequency components *L*, the noise subspace matrix $S_{noise}$ consisting $N_{sub}-L$ eigenvectors of $R_{yy}$ corresponding to the smallest $N_{sub}-L$ eigenvalues is obtained, and the super-resolution spectrum is obtained as
(53)$$sp\left(f\right)=\frac{1}{v^{H}\left(f\right)S_{noise}S_{noise}^{H}v\left(f\right)},$$
where the direction vector v(f) at a frequency *f* is defined by
(54)$$v\left(f\right)\left[n\right]=\exp\left(j2\pi fnT_{PRI}\right),\;n=0,1,2,...,N_{sub}-1,$$
In (Equation 54), *f* can be set with infinite resolution, which is a major advantage of the MUSIC algorithm. In this study, we divided the signal into respiratory and cardiac signals and set *f* around $f_{r}$ and $f_{c}$ with a very high resolution, which may require very long observations if the FT is employed.

## 4. Experimental Results

### 4.1. Experimental Condition

To demonstrate the efficiency of the proposed method for VSE in the presence of body movement using measurements from general low-cost short-range radar, we used both IR-UWB and FMCW radars. For the IR-UWB radar, we employed the parameters of the X4M03 produced by Novelda (Oslo, Norway), and for the FMCW radar, we used those of the Distance2GoL by Infineon (Neubiberg, Germany) (see Table 1). The radar signals of both radars were modeled, and $f_{r}$ and $f_{c}$ were estimated using the proposed method. A time frame of 40 s was used to obtain the spectrum with a 0.025 Hz resolution, and the time frame was moved with a step of 1 s to ensure overlapping neighboring frames.

Measurements were conducted using both the X4M03 and Distance2GoL radars. The subject was positioned approximately 2 m away from the radar sensor. Using the same frame time and frame step as in the experiment, the $f_{r}$s and $f_{c}$s of the received signal were estimated. To evaluate the accuracy of these estimations, actual $f_{r}$s and $f_{c}$s were measured using a wearable metabolic system, the K5 produced by COSMED, and compared with the estimated values (see Figure 9).

To facilitate computation, we used MATLAB for the experiment and measurement processes. We employed the functions unwrap(·), eig(·), and butter(·) for phase unwrapping, eigenvalue decomposition, and the Butterworth filter, respectively. For envelope detection, the parameters were set as L=150 and $▵t_{w}=$ 6 s for IR-UWB and L=300 and $▵t_{w}=$ 6 s for FMCW. The Butterworth filter parameters were defined as $10/60\leq f_{1}\leq 30/60$ and $55/60\leq f_{2}\leq 100/60$ for the ranges of RR and CR, respectively. For $K(t_{p})$, σ was set to 0.7. The MUSIC algorithm was applied to the zero-padded signal, with zeros equal to the length of the signal added; $M_{sub}$ was set to 801 for IR-UWB and 1601 for FMCW. For each radar, the frequency was sampled at 0.0033 Hz for RR and 0.005 Hz for CR.

In this study, we compared the estimation accuracy of the proposed method with that of four different previous methods cited in [16,20,25,35]. The methods are as follows:The method in [16] compensated for phase error by detecting constant Doppler shift due to random body movement.The conventional method in [20] used a fuzzy rule to mitigate the effects of random movement.The method in [25] employed empirical mode decomposition (EMD).The method in [35] used VMD to detect vital signs.

### 4.2. Estimation Accuracy and Robustness of the Proposed Scheme

The experimental results confirm the efficiency of the proposed method (Figure 10 and Figure 11). Entropy minimization successfully aligned the RPs of the IR-UWB and FMCW radars (Figure 10b and Figure 11b). However, the phase of the signal post-PCA was considerably distorted owing to motion, obscuring the sinusoidal patterns of the RR and CR (Figure 10c and Figure 11c).

The proposed MOCOM #1 effectively removed motion artifacts. As the movement was accurately modeled by (Equation 38), the phase error was substantially reduced, revealing a clear sinusoidal pattern in the phase components’ envelope (Figure 10d and Figure 11d). Additionally, MOCOM #2 further eliminated residual errors, resulting in an even more pronounced sinusoidal pattern (Figure 10e and Figure 11e).

After applying LPF and HPF, the phase data clearly represented RR and CR (Figure 10f,g and Figure 11f,g). Compared with the combined signal of RR and CR, the sinusoidal pattern of CR was distinctly observable. Ultimately, the MUSIC algorithm accurately estimated both RR and CR (Figure 10h,i and Figure 11h,i).

The robust VSE performance of the proposed method was validated by comparing the estimated performance of the proposed scheme with the four existing techniques, as shown in Figure 12 and Figure 13. Furthermore, to evaluate the accuracy of the proposed scheme quantitatively, we computed the root-mean-square error (RMSE) between the RR and CR measured by the wearable metabolic system and estimated by the two radar systems. The RMSE is calculated as
(55)$$\mathrm{RMSE}=\sqrt{\frac{1}{N_{m}}\sum_{i=1}^{N_{m}}{\left({\bar{f}}_{c}\left(i\right)-{\hat{f}}_{c}\left(i\right)\right)}^{2}}$$
where ${\bar{f}}_{c}$ and ${\hat{f}}_{c}$ are the measured and estimated values, respectively, and $N_{m}=75$ denotes the number of frames.

The results of the experiments conducted using the two radar systems are summarized in Table 2. Both the proposed and existing methods described in [35] demonstrated accurate estimation of RRs with an RMSE of less than 0.1 Hz in scenarios involving IR-UWB and FMCW radar systems. In contrast, conventional methods outlined in [16,20,25] exhibited significantly higher RMSEs when estimating RRs using the IR-UWB radar system. The performance of the proposed method in CR estimation was also significantly superior to that of the conventional methods.

In the Fuzzy rule-based method presented in [20], the initially estimated RRs and CRs were used as threshold values for the membership function in fuzzy logic rules. This method depends solely on fuzzy rules without incorporating any compensation techniques for VSE in subjects with random body movements. Consequently, the accuracy of VSE using this approach is highly dependent on the selected fuzzy logic thresholds. Determining appropriate thresholds using initial estimates is particularly challenging in this experiment, as vital signs were estimated for a subject with random body movements, even in the initial phases of the experiment.

Conventional methods referenced in [16,25,35] identified the desired RR and CR by selecting the first maximum peak in each spectrum. These peaks were isolated using bandpass filters in [16], EMD in [25], and VMD in [35]. However, because these methods do not account for motion compensation, their VSE accuracy is compromised by phase distortions caused by random body movements. Consequently, these conventional methods are inadequate for detecting vital signs in individuals with significant body movement.

## 5. Conclusions

In this study, we present a novel algorithm designed to enhance VSE accuracy in the presence of body movement by incorporating a motion compensation technique. The proposed scheme follows a four-step process: 1. Radar signal denoising using PCA: this step reduces noise and enhances signal clarity. 2. Phase distortion compensation with two MOCOM methods: These techniques correct distortions caused by body movements. 3. Separation of respiratory and cardiac signals using a bandpass filter: this step isolates the vital signs for a more accurate analysis. 4. VSE using the MUSIC method: this algorithm step ensures a precise estimation of vital signs.

Experimental results with IR-UWB and FMCW radar systems demonstrated that our method provides more accurate and robust VSE than conventional methods, especially in scenarios involving random body movements.

Future studies will focus on the biomechanical analysis of walking humans. We plan to extend our algorithm to estimate not only vital signs but also walking parameters. The presence of various micro-movements related to respiration, heartbeat, and limb movements leads to mutual interference and phase ambiguity in the received signals. To address these challenges, we aim to employ an MIMO radar system.

## Figures and Tables

**Figure 1 sensors-24-06765-f001:**
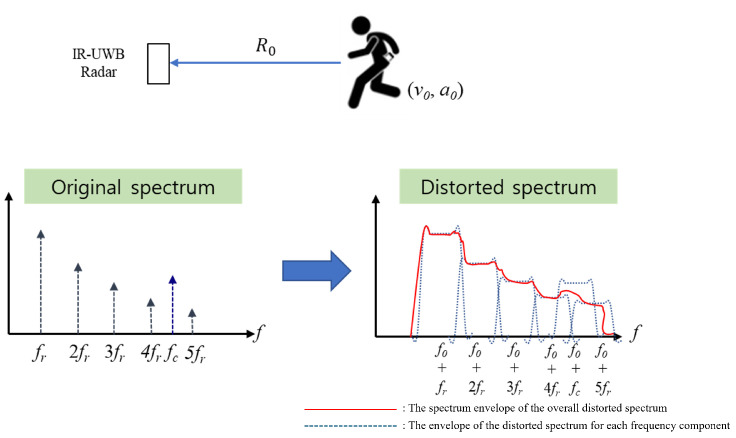
Example of the spectrum distortion caused by $v_{0}$ and $a_{0}$.

**Figure 2 sensors-24-06765-f002:**
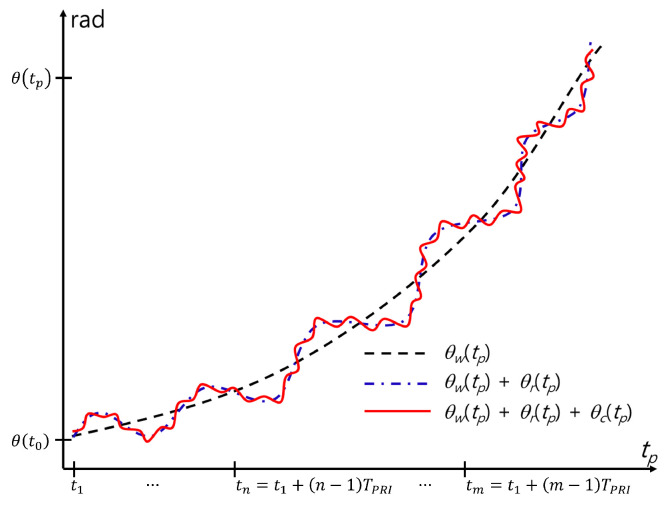
Example of the phase distortion caused by $v_{0}$ and $a_{0}$.

**Figure 3 sensors-24-06765-f003:**
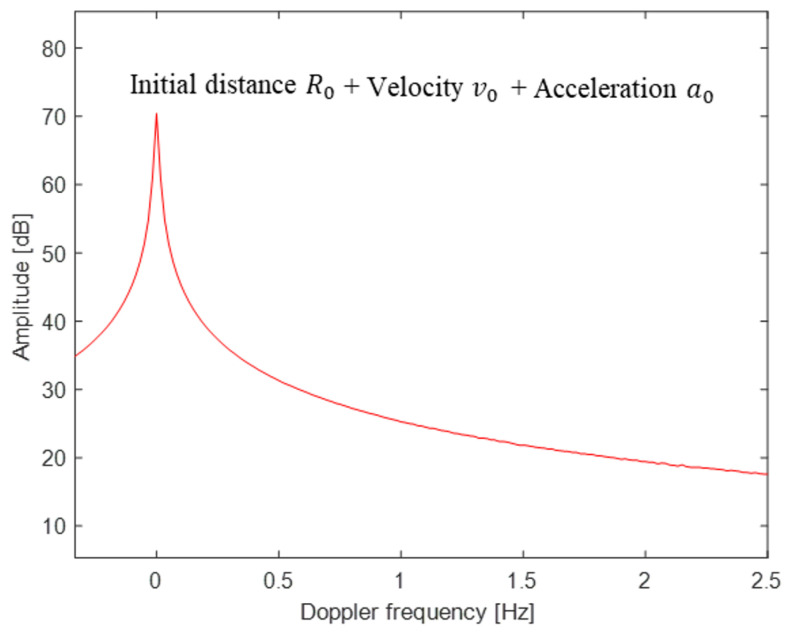
Example of blurred noise due to $v_{0}=0.01$ m/s and $a_{0}=0.0001$ ${m/s}^{2}$ (SNR=30 dB). Note that even very small velocities and accelerations can alter the spectrum of the phase.

**Figure 4 sensors-24-06765-f004:**
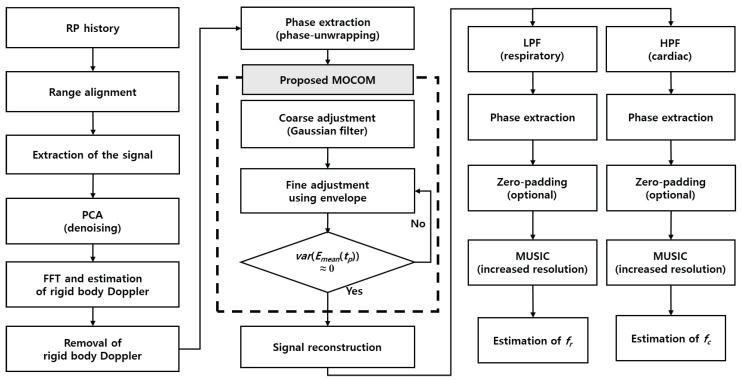
Proposed signal processing procedure.

**Figure 5 sensors-24-06765-f005:**
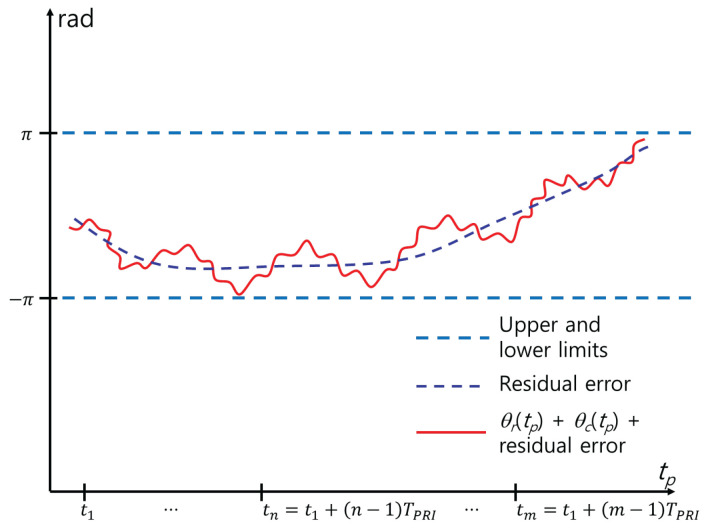
Phase in Figure 1 after coarse MOCOM #1 using (Equation 38).

**Figure 6 sensors-24-06765-f006:**
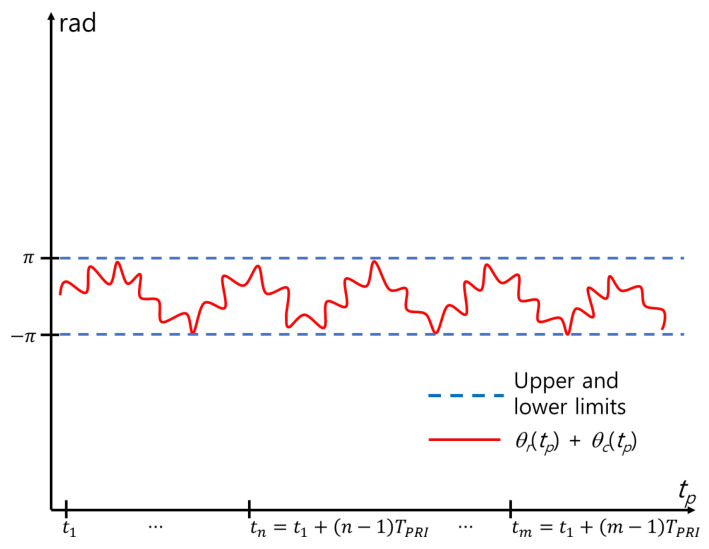
Phase after ideal MOCOM.

**Figure 7 sensors-24-06765-f007:**
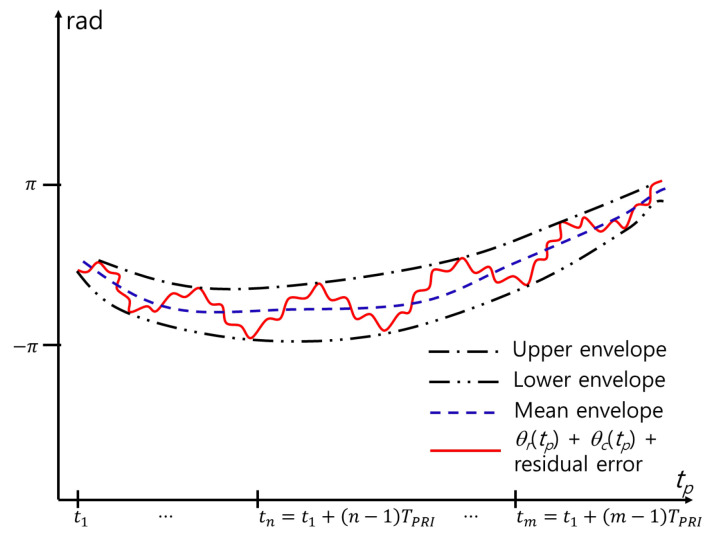
Envelopes for MOCOM #2.

**Figure 8 sensors-24-06765-f008:**
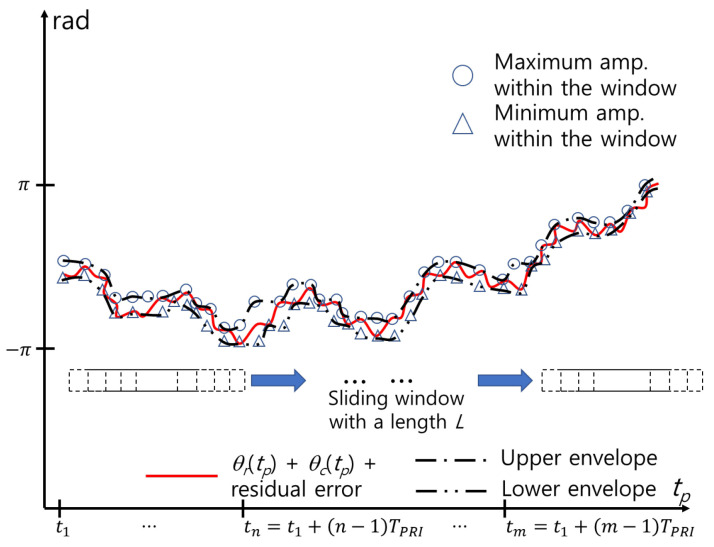
Estimation of envelopes using cubic spline interpolation.

**Figure 9 sensors-24-06765-f009:**
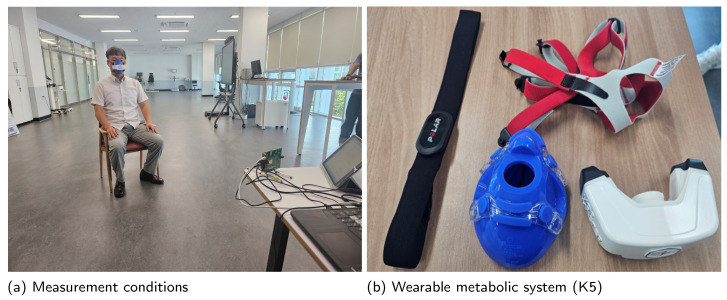
Measurement of VSE.

**Figure 10 sensors-24-06765-f010:**
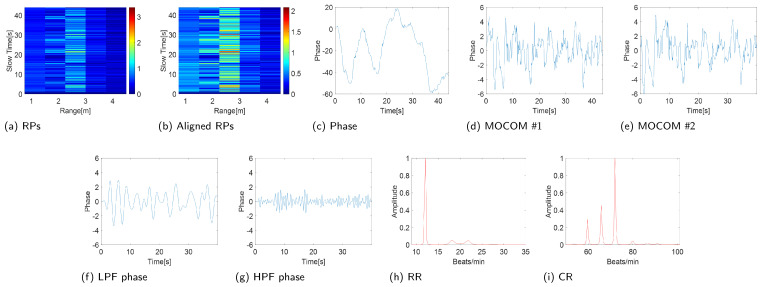
Experimental results (IR-UWB: X4M03).

**Figure 11 sensors-24-06765-f011:**
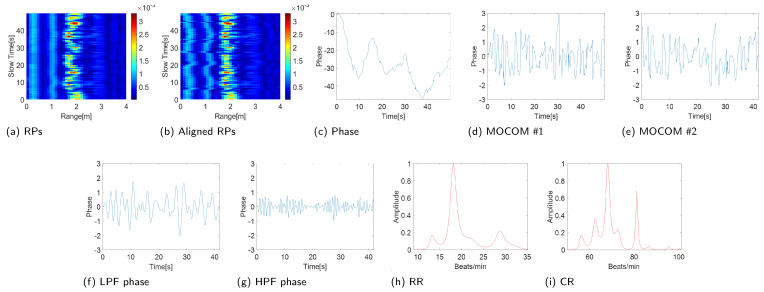
Experimental results (FMCW: Distance2GoL).

**Figure 12 sensors-24-06765-f012:**
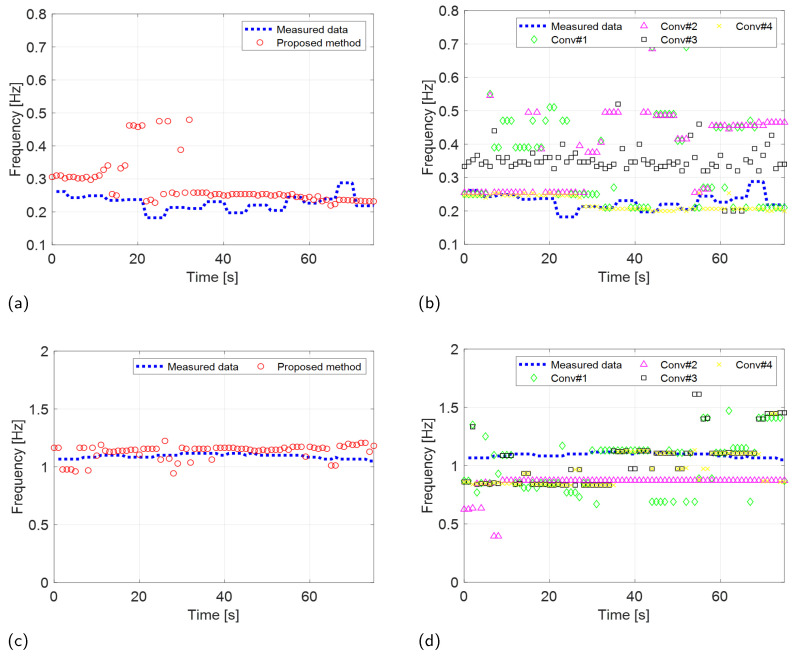
Experimental results (IR-UWB: X4M03) comparing the VSE performance of the proposed and conventional methods. RR estimation of the (**a**) proposed and (**b**) conventional methods. CR estimation of the (**c**) proposed and (**d**) conventional methods.

**Figure 13 sensors-24-06765-f013:**
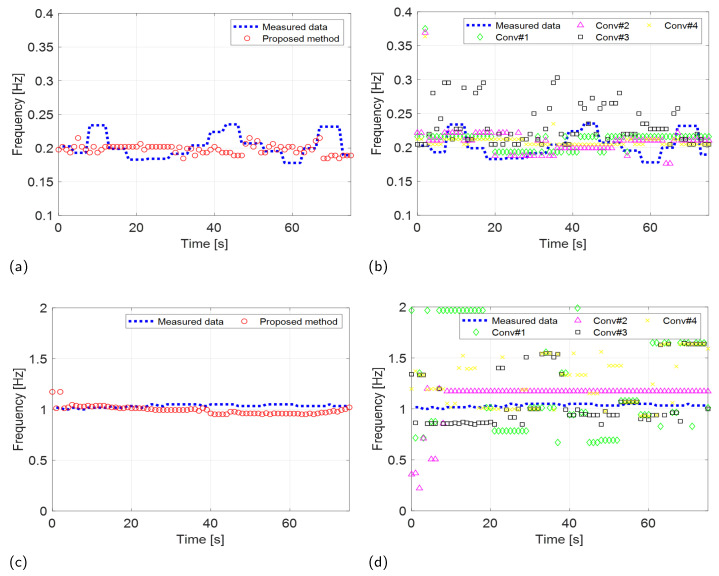
Experimental results (FMCW: Distance2GoL) comparing the VSE performance of the proposed and conventional methods. RR estimation of the (**a**) proposed and (**b**) conventional methods. CR estimation of the (**c**) proposed and (**d**) conventional methods.

**Table 1 sensors-24-06765-t001:** Experiments and measurement parameters.

Parameter	X4M03	Distance2GoL
Center frequency ($f_{cen}$)	7.29 GHz	24 GHz
Bandwidth (*B*)	1.5 GHz	200 MHz
Maximum range	10 m	10 m
Frame time ($T_{f}$)	40 s	40 s
Pulse repetition time ($T_{PRI}$)	0.0417 s	0.02 s
No. of samples per frame	960	2000
Frame time interval	1 s	1 s
Observation time	115 s	115 s
Distance to target	2 m	2 m

**Table 2 sensors-24-06765-t002:** Comparison between RMSE of estimated RR and CR of conventional methods and proposed method.

	Proposed	#1	#2	#3	#4
RR (IR-UWB)	0.091	0.2675	0.3129	0.1315	0.0298
CR (IR-UWB)	0.0772	0.2427	0.26	0.2203	0.1898
RR (FMCW)	0.0219	0.028	0.0288	0.043	0.0273
CR (FMCW)	0.064	0.5084	0.2102	0.2759	0.3319

## Data Availability

No new data were created or analyzed in this study. Data sharing is not applicable to this article.

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
