# Peer review of "Enhanced Vital Parameter Estimation Using Short-Range Radars with Advanced Motion Compensation and Super-Resolution Techniques"

_sensors, 2024, doi:10.3390/s24206765_

Round 1
Reviewer 1 Report
Comments and Suggestions for Authors
R266: The authors should explain what represent figure 5 (with residual errors).
In figure 5 are represented with red and blue only the residual values?
Reviewer 2 Report
Comments and Suggestions for Authors
The paper proposes a method for motion compensation to estimate the respiration rate RR and cardiac rate CR with super-resolution accuracy. The proposed method effectively models the radar signal phase and compensates for motion. Additionally, applying the super-resolution technique to RR and CR separately further increases the estimation accuracy. The paper incorporated de-noising, motion compensation, then applying superresolution. Experimental results from the IR-UWB and FMCW radars demonstrate that the proposed method successfully estimates RR and CR even in the presence of body movement. The following are necessary notes:
1-There is extensive mathematical analysis.
2-A number of techniques have been proposed, and implemented.
3- Figures 2, 5, 6, 7, and 8, are shown as graphical explanations, where there are no values on the two axes of each of them. It would be better to improve the presentations by showing typical values on the axes.
4-Experimental at a small range of 2 m are given for two types of radars.
Reviewer 3 Report
Comments and Suggestions for Authors
This paper proposes an efficient method incorporating MOCOM to estimate RR and CR with super-resolution accuracy. The proposed method effectively models the radar signal phase and compensates for motion. Applying the super- resolution technique to RR and CR separately further increases the estimation accuracy. Experimental results demonstrate that the proposed method successfully estimates RR and CR .
There are still several problems in the paper that need to be revised:
1. Each step in 3.1 should be briefly described based on what characteristics of the signal and noise select the corresponding step processing method!
2. What equipment is used to measure the measured data in Figure 12 and 13, please explain;
3. The existing part of the article on the specific treatment method is too long and should be brief.
Comments on the Quality of English LanguageThe paper clearly presents the content and results of the study
